# Questionnaire Survey on Driving among Patients with Age-Related Macular Degeneration in Japan

**DOI:** 10.3390/jcm10214845

**Published:** 2021-10-21

**Authors:** Chikako Hara, Miki Sawa, Fumi Gomi, Kohji Nishida

**Affiliations:** 1Department of Ophthalmology, Graduate School of Medicine, Osaka University, Osaka 5650871, Japan; sawamikimd@gmail.com (M.S.); fgomi@hyo-med.ac.jp (F.G.); knishida@ophthal.med.osaka-u.ac.jp (K.N.); 2Department of Ophthalmology, Sakai City Medical Centre, Osaka 5938304, Japan; 3Department of Ophthalmology, Hyogo College of Medicine, Nishinomiya 6638501, Japan

**Keywords:** age-related macular degeneration, driving, questionnaire, signal colour, elderly persons

## Abstract

Purpose: This study aimed to assess driving capabilities in patients with exudative age-related macular degeneration (AMD) causing unilateral blindness or paracentral scotoma without vision deterioration. Methods: Of the 275 patients with AMD who responded to a questionnaire regarding car driving at Osaka University Hospital, we excluded 78 patients who answered that they had never driven. Finally, 197 patients were included (50 with bilateral and 142 with unilateral AMD). We investigated the relationship between the questionnaire findings and best-corrected visual acuity (BCVA). Results: The mean age was 74.8 ± 6.9 years, and the mean BCVA in the right and left eyes were 0.48 and 0.47, respectively. A negative correlation was observed between the proportion of patients who stopped driving due to AMD and the vision in the worse eye (*p* < 0.0001); however, 66% of participants were still driving. Regardless of the BCVA, 84% of them wished to continue driving. Concerning perceived dangerous situations, all patients reported an oversight of people or signals and night driving; further, patients with unilateral and bilateral vision deterioration reported vision narrowness and difficulty with discerning signal colours, respectively. Conclusion: Despite the associated danger, patients with AMD continued driving. Close attention should be paid to the driving activities among patients with AMD, even if they have passed the relevant driving tests.

## 1. Introduction

Worldwide, age-related macular degeneration (AMD) is a leading cause of visual impairment in older people [1,2,3,4]; moreover, it causes central scotoma development and impairs macular functions, which reduce visual acuity and contrast sensitivity [5,6,7,8]. The increasingly ageing population will further increase the number of patients with AMD. Driving can be a crucial aspect of their daily life activities; moreover, many patients continue driving even during AMD treatment. Additionally, the introduction of anti-vascular endothelial growth factor drugs has led to an increase in the number of patients driving after AMD diagnosis [9]. Therefore, there is a need to determine driving-related risks among patients with AMD.

There have been several studies on car driving skills and related dangers in patients with eye diseases. For example, compared with healthy people, patients with glaucoma were observed to experience more inconvenience when driving [10,11]. They were likely to overlook stop signs and make operational errors in a driving simulator test [12,13]. Additionally, there is a high risk of car accidents while driving among patients with glaucoma, [14,15] retinitis pigmentosa, [16,17] and cataracts, which are common diseases [18] Unlike AMD, which involves a central scotoma, the aforementioned diseases could impair peripheral vision. Further, since AMD affects older persons, there is a need to consider the driving-related dangers based on the deterioration of their physical ability, which is unique to older persons.

This study aimed to characterise daily-life car driving in patients undergoing AMD treatment and the real-world perceived risks; moreover, we assessed the relationship between each risk and the visual acuity of both eyes.

## 2. Materials and Methods

The study was approved by the Ethics Committee of Osaka University Graduate School of Medicine (approval number 10039) and conducted according to the tenets of the Declaration of Helsinki. All patients consented to participate in the study. We included 275 consecutive patients (214 males and 61 female participants) treated for neovascular AMD in the Department of Ophthalmology, Osaka University Hospital, who received a questionnaire regarding car driving. Of the 275 patients, 197 (179 male and 18 female patients) who answered that they were driving before or during the study period were examined. The remaining 78 patients (sex, 35 male and 43 female patients) who answered that they had never driven were excluded from this study. The questionnaire items (Table 1) were as follows: (1) current driving status; (2) driving frequency; (3) whether the individual wanted to continue driving in the future; (4) whether the individual felt danger while driving; and (5) for individuals who answered ‘yes’ to the previous question, the circumstances in which they felt danger. We analysed the correlation between the questionnaire results and the best-corrected visual acuity (BCVA) using Landolt C charts for each eye at the time of answering the questionnaire. Further, we determined whether the patients had unilateral or bilateral AMD. Based on the individual BCVA of the first and second eyes, the patients were grouped as shown in Figure 1: Group 1, patients with a BCVA of ≥0.7 in the better eye and ≥0.5 in the other eye; Group 2, patients with a BCVA of ≥0.7 in the better eye and ≥0.1 and <0.5 in the other eye; Group 3, patients with a BCVA of ≥0.7 in the better eye <0.1 in the other eye; and Group 4, patients with a BCVA of <0.7 in both eyes. (Figure 1) In Japan, the driver’s license system entails attending a course and passing a visual acuity test to obtain and renew a car license once every 3–5 years. A BCVA of ≥0.7 in both eyes is required in the visual acuity test. If the visual acuity of one eye is <0.7, but the visual acuity of both eyes is ≥0.7, or if the visual acuity worsens after the renewal of the license, the license is valid. Therefore, we defined a BCVA value of 0.7 as good. Moreover, we defined a standard value of 0.5 for BCVA as poor, as it allows the performance of activities of daily living using both eyes. Additionally, a BCVA of 0.1 is defined as social blindness in the United States.

Statistical comparisons were made between the response of each group using chi-square or Fisher’s exact tests, and the tendency was analysed using the Cochran–Armitage tendency tests. For all analyses, a *p*-value < 0.05 was considered statistically significant.

## 3. Results

Among the 197 patients, 179 male and 18 female individuals were currently or previously driving (mean age: 74.8 ± 6.9 years). The mean BCVA values in the right and left eyes were 0.48 and 0.47, respectively. There were 50 patients with bilateral AMD, 139 with unilateral AMD with another healthy eye, and 8 with unilateral AMD with another diseased eye (epiretinal membrane, 5; glaucoma, 1; central retinal vein occlusion, 1; hypertension retinopathy, 1). Except for AMD, all the other diseases were in the chronic stage. Moreover, there were 48, 11, and 138 patients with bilateral pseudophakia, unilateral pseudophakia with another phakia, and bilateral phakic eyes, respectively. Although all phakic eyes had age-appropriate cataracts, no eyes had cataracts for which surgery is indicated to deteriorate visual acuity.

Among the included patients, 130 (66%) answered that they were still driving. All cases were classified into the BCVA-based four groups, as shown in Table 2. The patients in Groups 1 and 2 were significantly younger than those in Groups 3 and 4 (*p* = 0.0001). There was no among-group difference in the other ophthalmic diseases. In the presence of cataracts, there were significantly more eyes with pseudophakia in Group 4 with BCVA deterioration (*p* = 0.047). Additionally, a positive correlation was observed between the proportion of patients with bilateral AMD and the severity of BCVA deterioration. Furthermore, 76.5% of the patients in Group 4 had bilateral AMD. (*p* < 0.0001) Bilateral AMD include eyes with exudative AMD that are currently being treated with anti-VEGF injections and eyes that have previously been treated but are not currently being treated due to scarring and non-exudative AMD. There was no among-group difference for the other ophthalmic diseases.

The proportion of patients who answered that they stopped driving because of AMD was positively correlated with the severity of BCVA deterioration even in patients with a good BCVA of ≥0.7 (*p* < 0.0001; Figure 2). Further, approximately one-third of the patients with bilateral BCVA < 0.7 were still driving. (Figure 2) In all groups, patients who were still driving reported driving more than four times per week; moreover, visual acuity was not associated with driving frequency. (*p* = 0.934) This finding suggested that the driving frequency of patients who continued driving was not affected by declining vision (Figure 3A).

Moreover, in question 3–2 (whether they would continue driving in the future), ≥80% of patients in Groups 1 and 2 (48 out of 54 patients (89%), and 49 out of 58 patients (84%), respectively) and more than two-thirds of patients in Groups 3 and 4 (six out of 10 patients (60%) and six out of eight patients (75%), respectively), aimed to continue as long as they had valid licenses.(*p* = 0.107) Most of the patients (109 out of 130 patients (84%)) with AMD wanted to continue driving (Figure 3B).

Regarding question 3–3 (‘Have you ever felt in danger while driving?’), approximately 50% of patients in all groups (Group 1 (66%), Group 2 (57%), Group 3 (70%), Group 4 (63%)) answered ‘yes’ (Figure 3C). The specific situations perceived as most dangerous in all groups included failing to notice pedestrians, oncoming vehicles, and traffic signs (Group 1 (32%), Group 2 (41%), Group 3 (27%), Group 4 (57%); Figure 4A); vision difficulties when driving at night and glare (Group 1 (61%), Group 2 (59%), Group 3 (53%), Group 4 (62%); Figure 4B) Additionally, compared with other groups, more patients in Group 4 (bilateral BCVA of ≤0.7) answered that it was difficult to identify the signal colour (Fisher’s exact test; *p* = 0.001, Cochran-Armitage tendency test; *p* = 0.005; Figure 4C). Fewer patients in all groups answered changing lanes, and there was no significant difference between them. (Figure 4E) The patients with poor unilateral vision tended to present the items with a narrow vision field (when determining the vehicle spacing (Fisher exact test; *p* = 0.0068, Cochran-Armitage tendency test; *p* = 0.0009) and understanding the surroundings (Cochran-Armitage tendency test *p* = 0.046); Figure 4F,G).

## 4. Discussion

This study found that 66% of patients with AMD were still driving; moreover, >50% of them continued driving while considering it dangerous and still wanted to continue. Although there was a negative correlation between the number of patients who quit driving and visual acuity deterioration, 130 (66%) of the included patients answered that they were still driving. This ratio and trend were consistent with those reported by the MARINA and ANCHOR studies [9]. Recently, driving among older people have become a global concern. AMD affects older persons, and its prevalence is expected to increase; therefore, it may cause driving-related issues among older people. Although patients with AMD are often older adults, it is difficult to discriminate patients with AMD based on their appearance as they are not physically compromised and do not have dementia. Therefore, individuals close to the patients are unlikely to advise the patients to stop driving cars. Additionally, a bilateral visual examination cannot specify unilateral vision loss. In AMD with juxtafoveal small lesions, even with scotoma and metamorphopsia beside the centre, the fovea has normal anatomy and may show valuable data in patients’ visual acuity tests. According to statistics regarding automobile accidents, there is a higher proportion of single-vehicle accidents, frontal collisions, and encountered accidents among older adults aged >65 years than among those in other age groups [19]. This could be attributed to reduced attention to the surroundings because of ageing-related physical changes and delays in response to brake and steering operations of a car. A survey of drivers who were older adults reported a decrease in recognising objects even without ophthalmic diseases [20]. Compared with older adults without visual problems, patients with AMD are likely to present poor body movements, including poor physical balance during walking or wandering [21,22]. Although these previous findings were related to walking, these AMD-related impediments might promote delays in response to brake and steering operations while driving in ageing individuals. A study using driving simulators for patients with AMD showed that compared with healthy people, the patients would often take additional time to recognise signals, have delayed brake responses, did not speed up, drove over the centre line, and did not recognise their accidents [23,24].

In our study, many patients with AMD reported oversights of signs and pedestrians, as well as driving at night, as specific situations that felt dangerous. This could be attributed to focal defects in the visual field. Most patients in Group 4 reported difficulties in identifying signal colours, which could be attributed to damage of macular cone cells in macular diseases. Therefore, attention should be paid to discriminating colour among patients with bilateral AMD. A few patients reported experiencing danger when changing lanes or determining vehicle spacing. This could be attributed to AMD-induced central focal defects in the visual field contrary to diseases that impair the peripheral view field, such as glaucoma and retinal pigmentosa [10,11,12,13,14]. This indicates the differences in the necessary driving precautions depending on the disease type and unilateral/bilateral involvement.

The study limitations were as follows: (1) Only the answers from the questionnaire were evaluated; therefore, the driving skills were not actually assessed. (2) Only visual acuity data were examined, and the grade of focal defect in the visual field was not included in the examination. As this study was conducted to obtain opinions from as many people as possible in daily AMD practice, the visual field test could not be performed. (3) Only patients with AMD were evaluated and were not compared with healthy older individuals. (4) The male-to-female ratio is biased. This study included only 18 females (9.1%), as 70% of Japanese patients with AMD are men [25] and a limited number of older women drive. However, this study revealed specific situations wherein patients with AMD may feel in danger while driving and examined the differences between unilateral and bilateral AMD patients.

Moreover, aggressive AMD treatment may maintain good visual acuity. A recent study reported that driving vision (≥70 ETDRS letters) in 59% of patients with AMD at baseline could be sustained for four years by anti-VEGF treatment [26]. Therefore, patients might often continue to drive for long periods while treating AMD. Medical professionals should consider the specific risks during driving according to each patient’s condition, a measure which might help patients with AMD drive more safely.

## Figures and Tables

**Figure 1 jcm-10-04845-f001:**
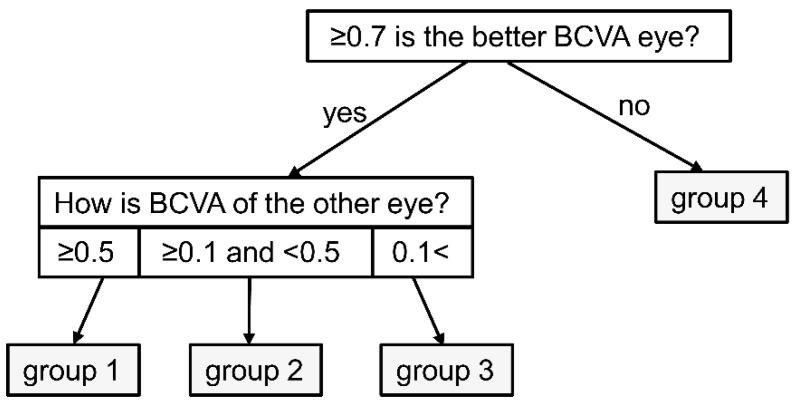
Flowchart of group classification.

**Figure 2 jcm-10-04845-f002:**
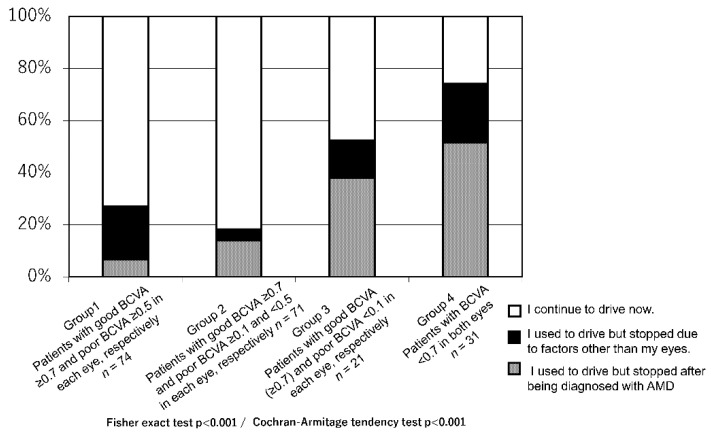
Breakdown of current and previous drivers.

**Figure 3 jcm-10-04845-f003:**
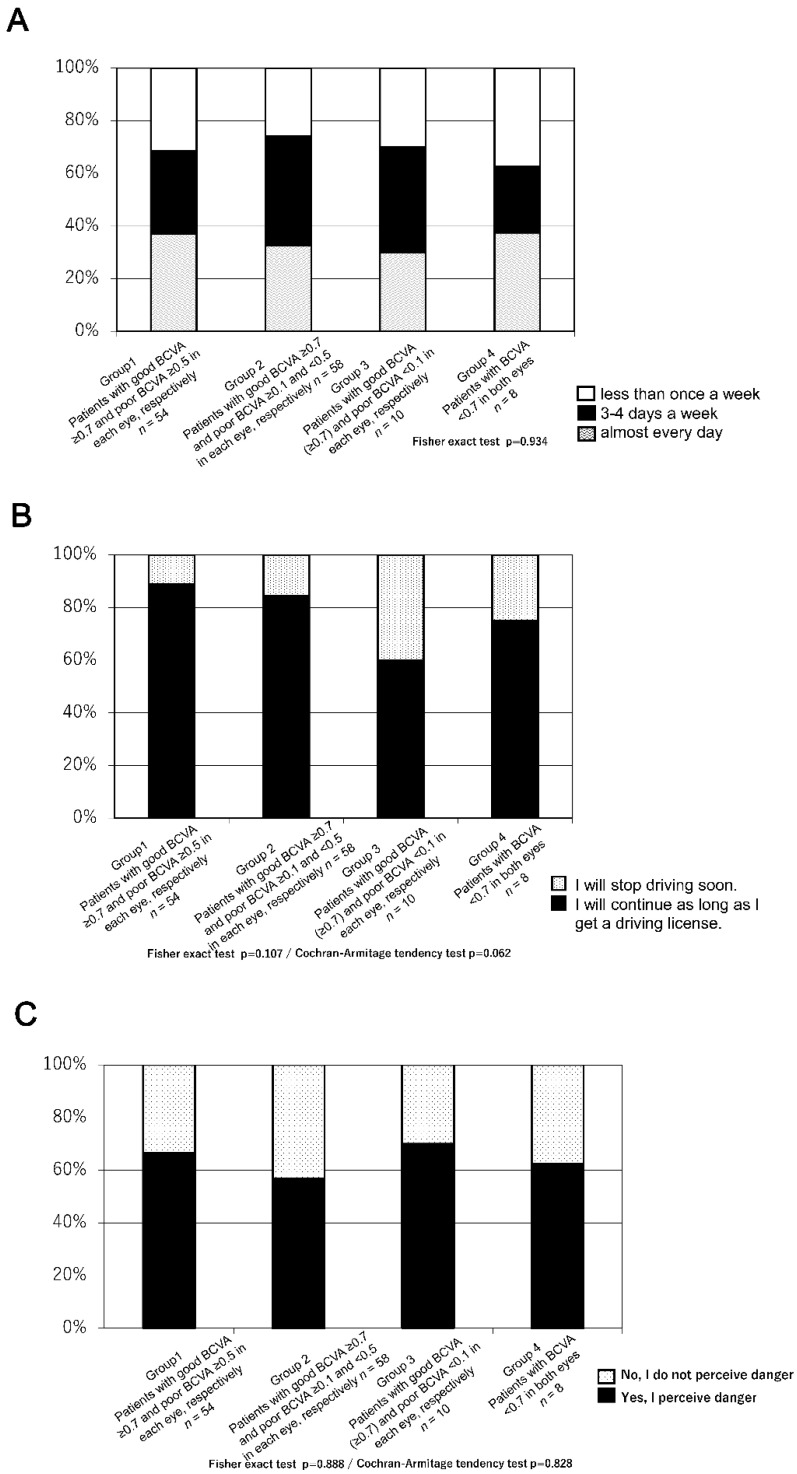
(**A**) Questionnaire results regarding the driving frequency among individuals who continue driving. (**B**) Questionnaire results on whether the patients would continue to drive. (**C**) Questionnaire results on perceiving danger while driving.

**Figure 4 jcm-10-04845-f004:**
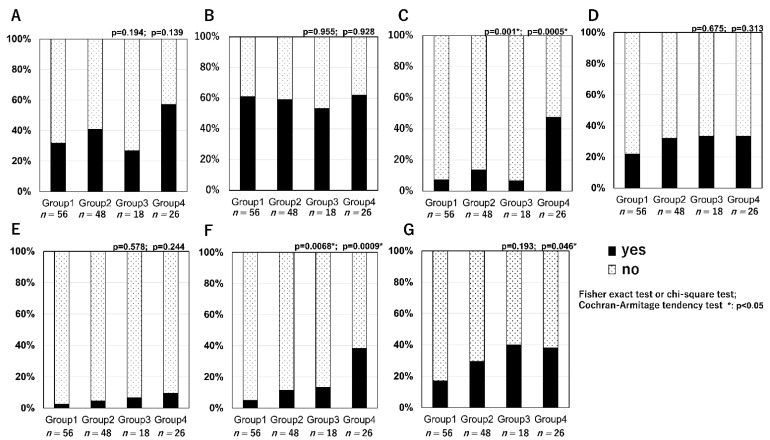
Specific situations wherein the patients with AMD felt in danger while driving. (**A**): failing to notice pedestrians, oncoming vehicles, and signs. (**B**): when driving at night (blur or glare). (**C**): discrimination of the signal colour. (**D**): when parking. (**E**): when changing lanes (**F**): when determining the vehicle spacing. (**G**): understanding the surroundings.

**Table 1 jcm-10-04845-t001:** Questionnaire contents.

Q1, Have you driven? A, yes; B, no
Q2, If the answer for Q1 is B (no), please choose one of the following: (A) I used to drive but stopped after being diagnosed with AMD. (B) I used to drive but stopped due to factors other than my eyes. (C) I have never driven
Q3, For individuals who answered A (yes) at Q1. Q1, How often do you drive? (A) almost every day; (B) 3–4 days a week; (C) less than once a week Q2, Will you continue to drive? (A) I will continue as long as I get a driving license; (B) I will stop driving soon Q3, Have you ever felt in danger while driving? (A) yes, I perceive danger; (B) no, I do not perceive danger.
Q4, For individuals who used to drive and have now stopped (answered A or B at Q2), as well as those who perceived danger when driving (answered A at Q3). Which situation have you felt in danger while driving? (multiple answers allowed) (1) failing to notice pedestrians, oncoming vehicles, and signs (2) when driving at night (blur or glare) (3) discrimination of the signal colour (4) when parking (5) when changing lanes (6) when determining the vehicle spacing (7) understanding the surroundings (8) others

**Table 2 jcm-10-04845-t002:** Characteristics of each group.

	Patients (*n*)	Years (Mean ± SD)	Eyes with Bilateral Pseudophakia: with Unilateral Pseudophakia with Another Phakia: with Bilateral Phakic Eye: (Ratio of Psedophakia)	Eyes with AMD Bilateral: Unilateral (Ratio of Binocular Cases)
Group 1 (Patients with good BCVA (≥0.7) and poor BCVA (≥0.5) in each eye, respectively)	74	74.1 ± 7.6	13:5:56 (20.9%)	2:72 (2.7%)
Group 2 (Patients with good BCVA (≥0.7) and poor BCVA (≥0.1 and <0.5) in each eye, respectively)	71	72.9 ± 6.3	12:5:54 (20.5%)	15:56 (21.1%)
Group 3 (Patients with good BCVA (≥0.7) and poor BCVA (<0.1) in each eye, respectively)	21	77.3 ± 6.1	6:1:14 (31.0%)	7:14 (33.3%)
Group 4 (Patients with BCVA < 0.7 in both eyes)	31	78.9 ± 6.8	17:0:14 (54.8%)	26:5 (83.9%)
*p*-value		*p* = 0.001 *	*p* = 0.047 *	*p* < 0.0001 *

AMD: age-related macular degeneration; BCVA: best-corrected visual acuity. * *p* < 0.05.

## Data Availability

Data available on request due to ethical restrictions. The data presented in this study are available on request from the corresponding author.

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
