# Peer review of "Questionnaire Survey on Driving among Patients with Age-Related Macular Degeneration in Japan"

_jcm, 2021, doi:10.3390/jcm10214845_

Round 1

Reviewer 1 Report

Dear authors, thank you very much for editing the manuscript. All major comments were answered, and statistical analysis and cataract status of the patients, were added. 

Author Response

We thank you and the reviewers for your thoughtful suggestions and insights. 

Reviewer 2 Report

In many countries, life expectancy is increasing and populations are aging. In order to maintain their independence, elderly people do not take a critical approach to their psychophysical abilities. The authors address a very important issue related to elderly drivers with an exudative form of age-related macular degeneration. This topic is not new, although it is certainly of interest to readers. For example, similar issues were presented by Baselius N et al. Driving vision in patients with neovascular AMD in anti-VEGF treatment Acta Ophthalmol 2021 Mar 5. doi: 10.1111/aos.14831, which the authors may wish to include in the discussion.

 I have the following comments:

  1. Line 103,104: “The proportion of patients who answered that they had stopped driving because of AMD was positively correlated with the severity of BCVA deterioration even in patients with a good BCVA of ≥ 0.7 (p<0.0001) (Fig. 2).”

Line 150, 151: “Although there was a negative correlation between the number of patients who quit driving and visual acuity deterioration, many patients continued driving.”

Summary:  “A negative correlation was observed between the proportion of patients who stopped driving due to AMD and the vision in the worse eye (p<0.0001);”

The above seems contradictory: the analysis once implies a negative and the other time a positive correlation. The authors should explain or correct it.

  1. Group 1 should be labelled more precisely:  line 67:  “patients with good BCVA (≥ 0.7) and poor BCVA (≥ 0.5) in each eye, respectively;”

however, BCVA  of ≥ 0.5 can be even 1.0 which cannot be considered as poor.

  1. What does “poor BCVA” mean?

In line 69 “poor BCVA” was described as  both ≥ 0.1 and < 0.5, while in line 70 “poor BCVA” was described as < 0.1.

Perhaps <0.1 should qualify as “very poor” if in the US it stands for “social blindness”

  1. 4 E was not described in the Results.
  2. In Fig. 4B group 3 (63%) has a bar lower than 60%. In Fig. 4B group 3 (63%) has a bar lower than 60%.
  3. Lines 53,59: the word ‘gender’ is generally more appropriate than ‘sex’.
  4. Grammatical errors, e.g. lines 185, 201
  5. Punctuation errors, misplaced periods. These errors appear multiple times in the text, the first time on line 35.
  6. Item 9 of the References should be amended.
  7. Line 101 says „bilateral AMD”. Do the authors mean bilateral exudative AMD treated with anti-VEGF injections?

Author Response

We thank you and the reviewers for your thoughtful suggestions and insights. 

In many countries, life expectancy is increasing and populations are aging. In order to maintain their independence, elderly people do not take a critical approach to their psychophysical abilities. The authors address a very important issue related to elderly drivers with an exudative form of age-related macular degeneration. This topic is not new, although it is certainly of interest to readers. For example, similar issues were presented by Baselius N et al. Driving vision in patients with neovascular AMD in anti-VEGF treatment Acta Ophthalmol 2021 Mar 5. doi: 10.1111/aos.14831, which the authors may wish to include in the discussion.

Response: We would like to thank the reviewer for the suggestion. Accordingly, we have cited reference number 26 in the Discussion section.

  1. Line 103,104: “The proportion of patients who answered that they had stopped driving because of AMD was positively correlated with the severity of BCVA deterioration even in patients with a good BCVA of ≥ 0.7 (p<0.0001) (Fig. 2).”

Line 150, 151: “Although there was a negative correlation between the number of patients who quit driving and visual acuity deterioration, many patients continued driving.”

Summary:  “A negative correlation was observed between the proportion of patients who stopped driving due to AMD and the vision in the worse eye (p<0.0001);”

The above seems contradictory: the analysis once implies a negative and the other time a positive correlation. The authors should explain or correct it.

Response: We would like to apologise to the reviewer for the confusion caused. Accordingly, we have corrected the expressions of lines 150 to 151 in the initial submission (Discussion section) as follows:

Line 157-159

Although there was a negative correlation between the number of patients who quit driving and visual acuity deterioration, 130 (66%) of the included patients answered that they were still driving.

  1. Group 1 should be labelled more precisely:  line 67:  “patients with good BCVA (≥ 0.7) and poor BCVA (≥ 0.5) in each eye, respectively;” however, BCVA  of ≥ 0.5 can be even 1.0 which cannot be considered as poor.

Response: We would like to apologise to the reviewer for the confusion caused. We used the term ‘poor BCVA’ to describe the visual acuity of the worse eye in the comparison of the visual acuity of both eyes. Accordingly, we have corrected the sentence as follows:

Line 67-72:

Based on the individual BCVA of the first and second eyes, the patients were grouped as shown in Fig.1: Group 1, patients with a BCVA of ≥0.7 in the better eye and ≥0.5 in the other eye; Group 2, patients with a BCVA of ≥0.7 in the better eye and ≥0.1 and <0.5 in the other eye; Group 3, patients with a BCVA of ≥0.7 in the better eye <0.1 in the other eye; and Group 4, patients with a BCVA of <0.7 in both eyes. (Fig.1)

  1. What does “poor BCVA” mean?

In line 69 “poor BCVA” was described as  both ≥ 0.1 and < 0.5, while in line 70 “poor BCVA” was described as < 0.1.

Perhaps <0.1 should qualify as “very poor” if in the US it stands for “social blindness”

Response: We would like to apologise to the reviewer for the confusion caused. As stated in our previous comment, we used the term ‘poor BCVA’ to describe the visual acuity of the worse eye in the comparison of the visual acuity of both eyes. Accordingly, we have corrected the sentence as follows:

Line 67-72:

Based on the individual BCVA of the first and second eyes, the patients were grouped as shown in Fig.1: Group 1, patients with a BCVA of ≥0.7 in the better eye and ≥0.5 in the other eye; Group 2, patients with a BCVA of ≥0.7 in the better eye and ≥0.1 and <0.5 in the other eye; Group 3, patients with a BCVA of ≥0.7 in the better eye <0.1 in the other eye; and Group 4, patients with a BCVA of <0.7 in both eyes. (Fig.1)

  1. 4 E was not described in the Results.

Response: We would like to thank the reviewer for the suggestion. Accordingly, we have added the following sentences to the Results section:

Line 130-131

Fewer patients in all groups answered changing lanes, and there was no significant difference between them. (Fig 4E)

  1. In Fig. 4B group 3 (63%) has a bar lower than 60%. In Fig. 4B group 3 (63%) has a bar lower than 60%.

Response: We would like to thank the reviewer for the suggestion. After checking, 63% of the statements were 53% wrong. Accordingly, we have corrected this as appropriate.

Line 125-127

vision difficulties when driving at night and glare [Group 1 (61%), Group 2 (59%), Group 3 (53%), Group 4 (62%)](Fig. 4B)

  1. Lines 53,59: the word ‘gender’ is generally more appropriate than ‘sex’.

Response: We would like to thank the reviewer for the suggestion. We have corrected this as appropriate.

Line 55

(gender, 214 males and 61 female participants)

  1. Grammatical errors, e.g. lines 185, 201

Response: We would like to thank the reviewer for the suggestion. We have corrected this accordingly.

Line 192-194

This indicates the differences in the necessary driving precautions depending on the disease type and unilateral/bilateral involvement.

Line 208-209

Therefore, patients might often continue to drive for long periods while treating AMD.

  1. Punctuation errors, misplaced periods. These errors appear multiple times in the text, the first time on line 35.

Response: We would like to thank the reviewer for the suggestion. We has revised the entire manuscript to correct these errors accordingly.

  1. Item 9 of the References should be amended.

Response: We would like to thank the reviewer for the suggestion. We have corrected this accordingly.

  1. Line 101 says „bilateral AMD”. Do the authors mean bilateral exudative AMD treated with anti-VEGF injections?

Response: Bilateral AMD include eyes with exudative AMD that are currently being treated with anti-VEGF injections and eyes that have previously been treated but are not currently being treated due to scarring and non-exudative AMD. We added this explanation to the manuscript as follows:

Line 102-105

Bilateral AMD include eyes with exudative AMD that are currently being treated with anti-VEGF injections and eyes that have previously been treated but are currently not being treated due to scarring and non-exudative AMD.

This manuscript is a resubmission of an earlier submission. The following is a list of the peer review reports and author responses from that submission.

Round 1

Reviewer 1 Report

Hara et al. presented a very interesting study on driving of AMD patients and if they feel dangerous. They show very interesting results that visual acuity does not seem to be related to how often patients drive, or if they preceive danger or not. 

In the methods I would suggest to improve line 63 to line 67. To me it is not clear how the groups are formed. Is group 1 visual acquity of one eye >0.7 and the other eye >0.5? 

Add a sentence that group 4 officially is not allowed to drive. And that the BCVA for a diving licence in Japan is 0.7 in one eye, not both. Or is it binocular visual acquity?

From reading this paragraph I thought visual acquity was in decimals. But later in the results section you say it is in LogMAR, please make clear what you did. 

there are only 18 females in the group. What would be the reason for that? Is there selection bias? Be very careful on any conclusions you make on the male:female ratio, because the ratio is skewed to begin with. So maybe leave out the third column of table 2. 

In line 95 you say "most of the patients", present the number and percentage here. 

Line 105 Remove this line. 

Figure 1 test statistical significance between groups. Show the significant differences with an *. 

Figure 2C. Add "yes, I perceive danger", "no, I do not perceive danger" to the legend of the chart. Just "yes/no" is somewhat confusing. 

Line 129/130: Add a sentence with the implication or consequence of no visual loss among patients with juxtafoveal lesions. 

In the last paragraph of the discussion add a clear conclusion of your paper. Also add a line on group 4 who is driving but not allowed to drive. 

Reviewer 2 Report

The topic of this article is very interesting and relevant. With respect to the original contribution of the study, the main limitation is the lack of statistical analysis and absence of evaluation of the cataract in participants of this study.

I recommend those revisions:

Line 37-43: Move to Discussion section

Line 56: Was the Questionnaire used in this study validated? Please add this information in the Methods section.

I understand that objective measurement of driving skill would not be possible. However, I recommend adding the information about the driving experience of the participants- how long do they drive in years and how do they rate their driving skills = subjectively. And more interestingly, if they have ever caused the car accident and when- before or after AMD diagnosis. 

Line 60: There is an absence of description of statistical tests used for correlation analysis. Please add it. It is crucial for other statements used in the Results section- “there was a correlation” or “there was no correlation”. If it was not statistically analyzed, then it must not be used in those statements. It should be stated as…”there is a tendency for…” and the informative value of the whole article is then poor.

Line 63-67: Please, describe better BCVA in the groups. What does it mean in each eye? Do you mean that for group 1 the BCVA of a better eye was higher or better than 0.7 and higher or better than 0.5 of a worse eye? Which was then the best BCVA of the worst eye - 0.7? It is not clear.

Line 67: Please add more details about legal conditions for obtaining or renewing the driving licence in Japan. It is different in different countries.

Line 74: Please add the BCVA of both eyes in each of the 4 groups (I recommend adding it to Table 2).

Line 78: Please add information about cataract in all eyes of all participants. It is crucial for BCVA. Not only AMD but cataract can severely affect the BCVA and it is not clear from the text if all of the participants were operated on cataract before being involved in this study. If not, I find the informative value of this article very poor.

Line 102: Do I understand well that patients from the Group 4 should not drive because of their poor visual acuity not suitable with the standard value of BCVA for driving in Japan? But there are quite a lot of them driving. Please, explain it.

Line 107: Number of participants in groups is not comparable. There are much less patients included in the Group 3 and 4. I recommend adding them to reach a comparable number of patients.

Line 132: Please add more citations for the statement that car accidents in elderly patients are more frequent. 

Line 146-148: Why was not done a visual field test and color test for participants? I find it very useful. Please, explain it more widely, not only “it was not done”- add the information WHY?